# Dateless Dendroarchaeology

**Ronald H. Towner**

Laboratory of Tree-Ring Research, The University of Arizona, Tucson, AZ 85721, USA; rht@email.arizona.edu

**Abstract:** The strength of dendrochronology is chronology. No other non-textual dating technique in the world provides the precision, accuracy, and resolution of dendrochronology. Indeed, dendrochronology is famous for dating prehistoric ruins, and Douglass' "Bridging the Gap" is still considered one of the greatest achievements in archaeology anywhere, but what happens when samples don't date? Should they simply be discarded as useless, stored until better chronologies and new techniques are available, or do they contain useful information for current research interests? Using undated collections from the southwestern US and northwestern Mexico, this paper discusses a variety of behavioral and environmental information present in samples, even if they cannot contribute to our chronological knowledge.

**Keywords:** dendroarchaeology; tree rings; Sonora; Texas

## 1. Introduction

Dendroarchaeological samples are products of both past tree growth and past human behaviors and contain three types of information: chronological, behavioral, and environmental [1]. Each of these types of information is entirely independent of the others, although they are certainly most powerful when used together. Unlike other dendrochronological samples collected for ecological or climatological research, however, dendroarchaeological samples are not only chosen by the researcher, but also, more importantly, chosen by past people for specific purposes. Thus, not all samples yield dates because they were not selected for their dating properties. The three types of information retained by archaeological tree ring samples are independent of each other, however. In other words, the absence of absolute Christian-calendar dates does not preclude delineating past human behaviors or illuminating aspects of past environmental conditions from dendroarchaeological samples. This paper uses examples from the US Southwest and Mexican Northwest to illustrate independent information gleaned from collections that yielded no absolute chronological information. These collections (Figure 1) were all analyzed using the Douglas method of skeleton plotting [2] and some ring series were measured as well.

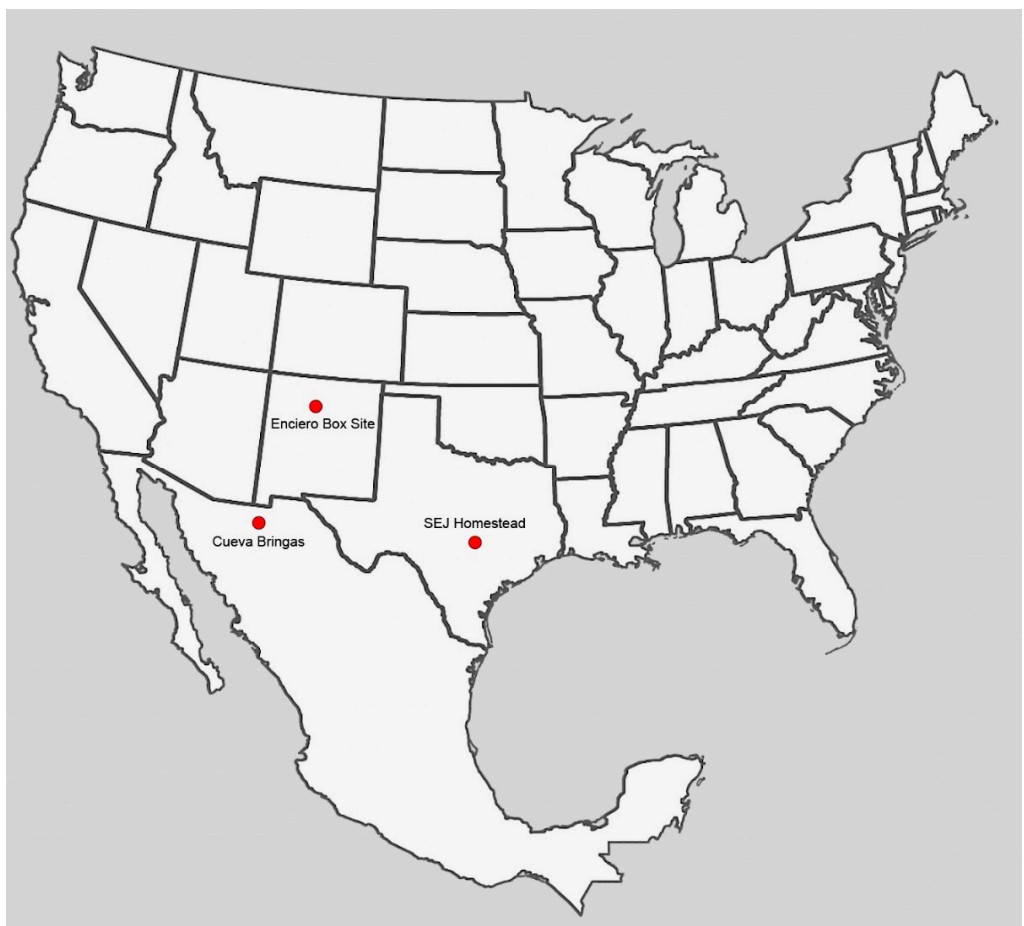

**Figure 1.** Maps of site locations discussed in the text.

## 2. Relative Chronological Information: Seasonality and Absolute Contemporaneity

Chronological information is usually considered in terms of absolute annual Christian-calendar dates. Tree ring samples that retain the last ring grown by the tree, however, also provide data concerning the season in which the tree was harvested. These seasonality data are more precise than annual resolution, even in the absence of a calendar date.

One excellent example of seasonality and absolute structure contemporaneity is derived from the Enciero Box site, a small multi-room site of unknown cultural affiliation in the Jemez Mountains of New Mexico (see Figure 1). The site consists of a room block and four ramada-like wooden structures built of Douglas-fir (*Pseudotsuga menziesii* (Mirbel) Franco) against a canyon wall adjacent to a narrow floodplain. One gray ware sherd of possible early Navajo manufacture (1600–1899 CE) is the only artifact present. Two of the four ramadas contain significant chronological information, even though the structures failed to date because the samples contain too few rings for matching against any master chronology; trees in the Southwest typically require 50 or more rings for confident pattern matching, but these samples retain fewer than 15 rings each.

Samples collected from the structures exhibit metal axe-cut beam ends (Figure 2), indicating they were cut sometime after 1600 CE when Spaniards introduced metal tools into the region. Two of the samples (Figure 3) cross-date with each other—the ring patterns match exactly, including placement of two narrow rings and a possible fire-scarred ring near the pith of both samples; thus, these two independent samples from different structures were cut in the same year, but that year remains unknown. Finally, the terminal rings on these samples exhibit early wood growth, indicating tree harvesting during the Douglas-fir growing season, probably between early May and late July in the Jemez area. Thus, the dendroarchaeological samples from the Enciero Box site indicate construction

of two small ramada-like structures during the Douglas-fir growing season in some year after 1600 CE. During precisely which post-1600 growing season the trees were harvested remains unknown.

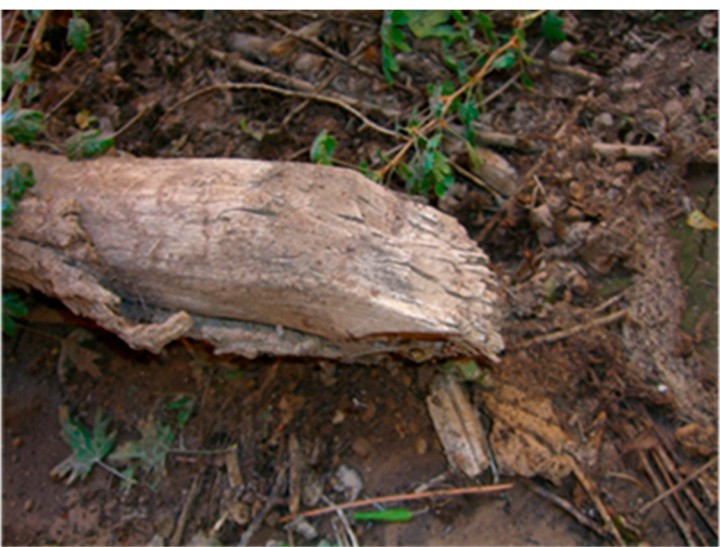

**Figure 2.** Metal axe-cut beam from the Enciero Box site.

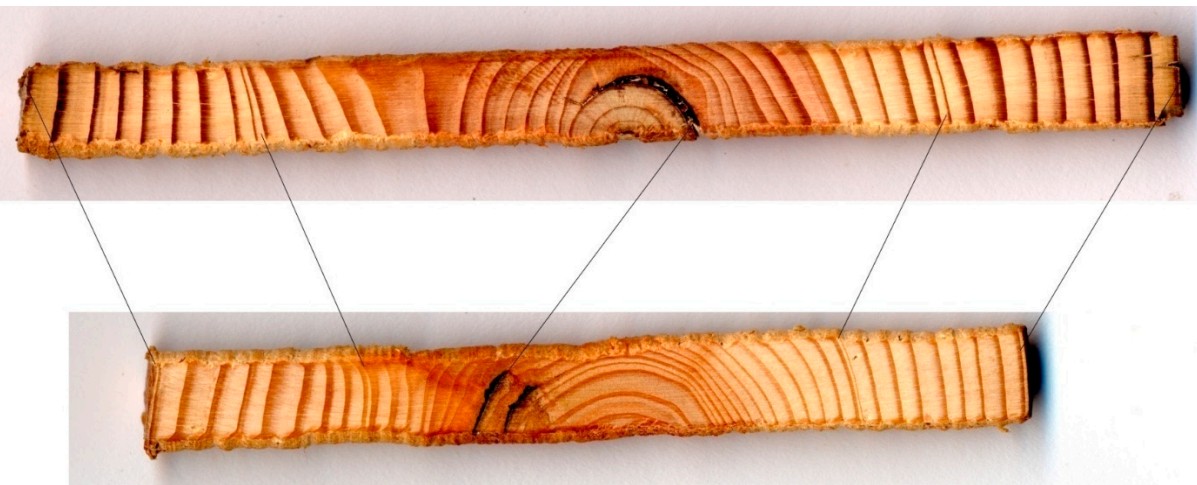

**Figure 3.** Ring attributes of Enciero Box site timbers.

## 3. Behavioral Information: Timber Sourcing and Procurement

Identifying the source of timbers used in archaeological sites is an important aspect of delineating the social and economic organization of the groups who procured trees and built structures. Timber sourcing can be accomplished in a variety of ways, from simply comparing ring-growth patterns to examining various isotopes fixed in the tree-rings indicative of specific growing locales [3]. The former has been used in Europe to identify oak timbers used in Viking sailing vessels [4,5] and to postulate the origins of Stradivarius violins [6]; it is currently being used to delineate the sources of timbers used in Egyptian tombs [7]. In the Southwest, nonlocal timber is relatively uncommon, but Betancourt et al. [8] suggested Chaco timbers were imported from distant sources based on their scarcity in local areas. Later, other researchers [9,10] verified Chacoan timber sources using strontium isotope analysis, and, more recently, Guiterman et al. [11] used ring-series measurements. Combined with an abundance of dated beams, these data have provided important information concerning the Chacoan economy and sphere of influence.

The example of timber source identification illuminated in this paper combines species distributions, ring-width patterns, and historical research. The Samuel Ealy Johnson (SEJ) house is a board-and-batten structure in LBJ National Historical Park in the Hill Country of Texas. The house belonged to Lyndon Baines Johnson's (the 36th US President) grandfather Samuel and was sampled to help determine when it was built, although historical references and oral histories suggested it was in existence by the early 20th century. The boards used to construct the SEJ house were 2.5 × 25 cm boards, often as long as 4.5 m (18 ft), and the batten pieces were probably cut from the same timbers. A total of 79 samples was collected from the SEJ house. There was some internal cross-dating—samples matched other samples—but none of the samples cross-dated against known chronologies either graphically or statistically. Two samples, adjacent boards in the structure, had such a high ring-series correlation that they are undoubtedly from the same tree. All of the samples are *Pinus palustris* Mill., a pine species that does not grow in the area today and probably never has been present. In the 19th century, there was a small pine forest east in Austin, TX [12], approximately 100 km (60 mi) to the east, but it is unknown whether those trees were *P. palustris* or another species. The current distribution of *Pinus palustris* (Figure 4) is restricted to the southeastern US. Importantly, the top end of one of the boards in the loft area was stamped "Calusa Lumber Co.". The company operated a logging and lumber operation on the Sabine River, which separates Texas and Louisiana, during the late 19th and early 20th centuries. Thus, it appears that Samuel Ealy Johnson acquired a load of lumber needing to be transported by ship down the Sabine river to a Texas port (probably Houston), then by railroad to San Antonio, and, finally, by wagon the last 50 km (30 mi) from San Antonio to Johnson City. This acquisition, therefore, informs us about the economic and transportation systems operating in late 19th century Texas. It also suggests that the Johnson family was more affluent than other homesteaders in the area who used local post oak (*Quercus stellata* Wangenh.) timbers in their construction [13].

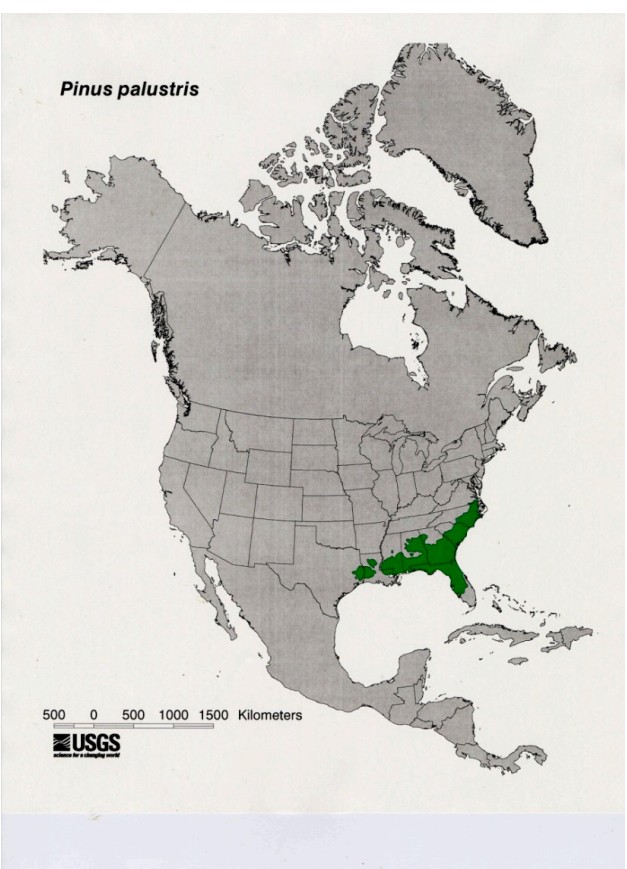

**Figure 4.** Modern distribution of *Pinus palustris* Mill. Map from the US Geological Survey.

*Procurement Techniques*

Timbers used by past peoples were procured using a variety of techniques: metal- or stone-axe; a variety of saw types; burning, breaking, recycling from existing structures; or collecting dead wood. The use of dead wood—trees that died a natural death and were then harvested—is common in some archaeological contexts. Dead wood is harder than living wood; therefore, stone axes generally were not used to procure such timbers [14]. This might suggest more dead wood use in the Southwest after the introduction of metal tools by the Spaniards. Such is not always the case, however, because Spanish, Anglo, and some Puebloan architecture required long, straight timbers to span large rooms.

Dead wood is often identified dendrochronologically by "++" dates, which provide terminus post quem chronological information [1,15]. Identifying dead wood use in the absence of chronological information is dependent on field observations. The most secure way to identify timbers procured as dead wood is by the presence of "root flares", the ends of timbers that retain portions of tree roots that must have been pulled out of the ground.

Breaking is another method of procuring wood for architectural elements, fuelwood, and artifact manufacture. An excellent example of the impacts that procurement by breaking elements have on date distributions is seen in Fremont granaries from Range Creek, Utah [16]. In that case, fewer than 10 percent of the 200 samples collected from maize storage structures yielded dates despite the use of normally dateable species, such as Douglas-fir (*Pseudotsuga menziesii*). These small structures required only small (<10 cm diameter) wooden elements that usually contained too few rings for cross-dating. These elements were undoubtedly dead branches procured by breaking.

## 4. Environmental Information: Growth Patterns and Species Selection

Environmental information derived from dendroarchaeological samples has typically taken the form of precipitation reconstructions that chronicle the variability of ring widths or correlate that ring-width variability with the climatic signal of interest. Indeed, much of modern-day dendrochronology centers around dendroclimatology—the retrodiction of temperature or precipitation to periods prior to written records. Such studies are possible because trees are natural archives of information, are typically long-lived organisms, and do not move around the landscape. In general, ring-width is controlled by precipitation in lower-forest-border trees and temperature at upper-forest-border trees; ring density, particularly latewood density, may also be a measure of temperature variability. All such reconstructions, however, rely on dated specimens used in conjunction with documentary data to calibrate the ring-grown climate variable relationship [17]). It is important here to note the seasonality of precipitation. Only recently have researchers been able to retrodict summer (warm) season precipitation patterns using measured portions of rings [8].

An important example of climatic information in the absence of dated specimens comes from cliff dwellings in the Sierra Madre of northern Sonora, Mexico. Cueva Bringas is a 20+ room cliff dwelling in the Rio Taraises drainage of Sonora that, based on limited ceramic and architectural data, was occupied sometime during the Formative Period, probably in the 14th or 15th century [18]. A total of 68 dendroarchaeological samples was collected from the site; most are Arizona cypress (*Cupressus arizonica* Greene), Chihuahua pine (*Pinus leiophylla* var. *chihuahuana* (Engelmann) Shaw), or juniper (*Juniperus* spp.). Unfortunately, the samples failed to date because they retained too few rings for cross-dating, contained abundant false rings, or exhibited erratic growth patterns.

Figure 5 shows a dendroarchaeological core from an Arizona cypress roof timber in Cueva Bringas. The sample retains only about 15 rings and was obviously a relatively fast-growing tree. Of concern here, however, is the presence of false rings, or false inter-annual bands, also known as inter-annual density fluctuations (IADF) [19] in almost every ring on the sample. Such inter-annual bands are caused by climatic factors related to the annual distribution of precipitation. Most trees begin growing in the spring when winter soil moisture is available for use; such ring growth includes large, thin-walled tracheids, known as earlywood cells. As summer approaches, soil moisture declines, and, if trees begin to

experience water stress, tracheids are produced that are smaller with thicker walls, known as latewood cells. If climatic conditions improve, the tree may again produce earlywood cells until near the end of the growing season. Recent research ([17,20] indicates that in much of the Southwest, false rings can be used to measure the intensity and duration of the North American monsoon—a climatic phenomenon that brings summer rains to much of northern Mexico and the bordering US states. The Cueva Bringas samples, therefore, suggest that such a monsoonal system has been operating in the Sierra Madre Occidental for hundreds of years, even though we cannot date the samples themselves.

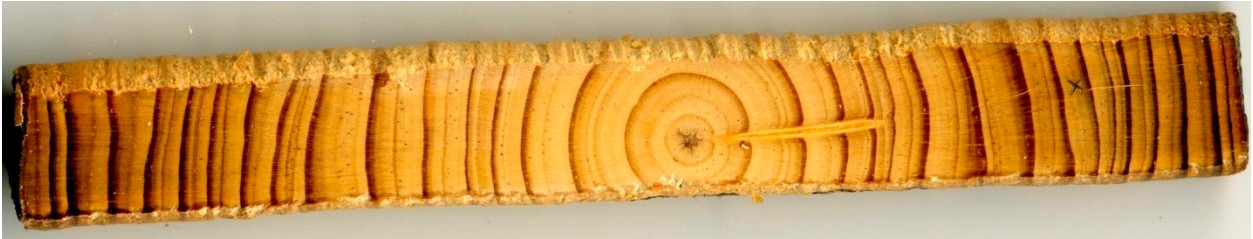

**Figure 5.** Cueva Bringas sample showing false rings.

## 5. Conclusions

The strength of dendrochronology has been, and always will be, the production of absolute calendar year dates, but, because archaeological samples were chosen by past people to meet their needs for structures, fuel, and artifacts, many such samples do not yield dates. For various reasons, such as microclimatic factors, small size, or undatable species, their ring series do not match the overall patterns evidenced in other trees. It is important, therefore, that archaeologists and dendrochronologists exploit all the information available in such samples. By examining the behavioral, ecological, and climatological information inherent in archaeological tree ring samples, we can gain additional insights into our shared human past, even in the absence of absolute calendar dates.

**Funding:** The work was partially funded by the US National Science Foundation (BCS-1923925) to which I am grateful.

**Institutional Review Board Statement:** No human subjects involved. No institutional approval required.

**Informed Consent Statement:** Not applicable.

**Data Availability Statement:** Study generated no new data.

**Conflicts of Interest:** The author declares no conflict of interest.

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
