# Peer review of "Dateless Dendroarchaeology"

_forests, doi:10.3390/f13020281_

Round 1

Reviewer 1 Report

Archaeological excavations frequently yield numerous samples of wood fragments which are very short in a dendrochronological sense. This paper presents a very interesting collection of examples of what can be done with such samples and what kind of information can be gleaned from them. Yet there are a couple of points of minor importance which the author may wish to consider.

Line 25: “samples are independent of each other. however” - comma

Fig 1. Three sites are mentioned in the text and depicted in Fig. 1. It seems that everyone can tell Texas from Mexico but is not it better to put site’s names on the map? Or mark them with different symbols and add a legend to the figure captions?

Line 45: “four ramada-like wooden structures” – it is getting clear from the subsequent discussion that wooden structures are made from Douglas-fir. But it is probably better to introduce tree species (along with its Latin name) in line 45.

Line 99 and Fig. 4 – citation of the source is needed

Line 157 – Juniperus from the capital letter

Line 162: “false rings, or false interannual bands” – these anatomical features are widely discussed in the literature (see, for example, Battipaglia et al., 2016. Structure and function of intra–annual density fluctuations: mind the gaps. Frontiers in Plant Science 7: 595) and it seems that generally accepted consensus has been established to call such features intra–annual density fluctuations or IADFs. Maybe it is better to at least mention this term as well.

Author Response

I have made all the requested corrections.

Two items I did not address. 

  1. There are figure captions with the figures, so I do not know what the reviewer intended.
  2. I am a native English speaker for 60+ years and have published in English for nearly 40 years. Because the reviewers provides no examples of improper grammar or word usage, I assume this was simply a mistake in checking the box on the form.

Reviewer 2 Report

General comment

The manuscript presents an interesting perspective on dateless dendroarcheological samples, most of them due to the low number of tree-rings that precludes dating using dendrochronological techniques. The main message of the manuscript is that, even if is not possible to date archaeological wood, they should not be discarded, and the maximum information needs to be recorded. One thing the author does not specifically mention is the importance of wood identification per se, that can give clues on the wood source.

I would suggest a new title: “Dateless Dendroarchaeology: three case studies from US Southwest and Mexican Northwest”

Specific comments

Line 44: Can you provide an image of the archaeological site? It would visually enrich the manuscript.

Line 68: I suggest changing the title to “Behavioral information: Timber source and procurement”

Line 84: Can you provide an image of the Samuel Ealy Johnson House?

Line 152: Can you add an image of the Cueva Bringas?

Author Response

I made the only text modification and change the subtitle on the Behavioral Information section.

Although additional figures would certainly enhance the paper, my funding would not cover the additional expense of more photographs, so I would rather not add the suggested figures.